# Probing the "Creativity" of Large Language Models: Can Models Produce Divergent Semantic Association?

**Honghua Chen** and **Nai Ding**[*]

Key Laboratory for Biomedical Engineering of Ministry of Education,
College of Biomedical Engineering and Instrument Sciences,
Zhejiang University / Hangzhou, China
{honghuachen,ding_nai}@zju.edu.cn

## Abstract

Large language models possess remarkable capacity for processing language, but it remains unclear whether these models can further generate creative content. The present study aims to investigate the creative thinking of large language models through a cognitive perspective. We utilize the divergent association task (DAT), an objective measurement of creativity that asks models to generate unrelated words and calculates the semantic distance between them. We compare the results across different models and decoding strategies. Our findings indicate that: (1) When using the greedy search strategy, GPT-4 outperforms 96% of humans, while GPT-3.5-turbo exceeds the average human level. (2) Stochastic sampling and temperature scaling are effective to obtain higher DAT scores for models except GPT-4, but face a trade-off between creativity and stability. These results imply that advanced large language models have divergent semantic associations, which is a fundamental process underlying creativity.[1]

## 1 Introduction

Large language models (LLMs) have exhibited unparalleled mastery of natural language (Bubeck et al., 2023). The primary capacity of producing the most probable next word is broadly generalizable to many language tasks, suggesting underlying cognitive abilities beyond specialized linguistic rules and patterns. There is observation that LLMs may possess reasoning abilities which is a core aspect of intelligence, including decision-making (Binz and Schulz, 2023) and theory of mind (Moghaddam and Honey, 2023). Meanwhile, there is also increasing interest in exploring LLMs' creativity, which is closely related to intelligence (Frith et al., 2021). Creative use of language, such as metaphor and

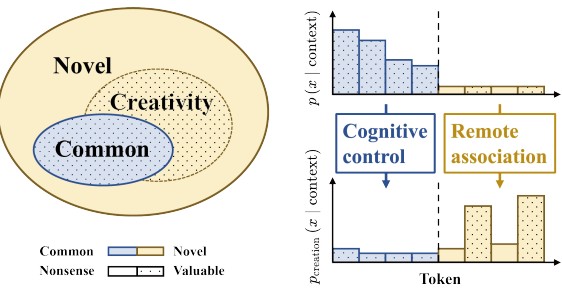

Figure 1: Creativity from the perspective of language distribution. Creative thoughts need to be novel and valuable, which need cognitive control to inhibit common tokens and remote association to find valuable tokens.

humor, is important during communication. OpenAI (2023) has reported GPT-4's ability to understand jokes, while subsequent works show limited capacity for LLMs to generate or explain humor (Jentzsch and Kersting, 2023; Hessel et al., 2023). As creativity is essential to the development of art, science, and everyday life for human (Gabora and Kaufman, 2010), it is non-trivial if models could produce creative content. Regarding to the curse of recursion for LLMs that training on generated data makes models collapse, one promising solution might be the novel language distribution through creative generation (Shumailov et al., 2023). But since LLMs represent word meaning and predict the next word in context, it seems paradoxical that such models could create ideas not seen in training. Here, we empirically investigate the creativity of LLMs by examining models' ability to generate divergent concepts.

A general definition of creativity is the ability to create something both novel and valuable (Runco and Jaeger, 2012). According to the dual-process theory of creativity (Beaty et al., 2014), creative thinking relies on remote association while inhibiting common ideas (Figure 1). Because of the intrinsic complexity, creativity is universally accepted to be unique to human beings, while models are

---

[*]Corresponding author

[1]We release our code at 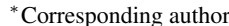 https://github.com/DingNLab/probing_creativity

regarded as great predictors to master the existing distribution but not qualified creators to generate new distribution. However, there have been evidences of models possessing creativity in different domains. For artistic creation, deep generative models reveals exquisite painting skills (Ramesh et al., 2022). For algorithms, models are able to generate original and superhuman strategies in board game (Silver et al., 2016).

As for language models, creativity is an emerging concern. Since GPT-2 (Radford et al., 2019), language models are able to naturally produce answers given a prompt, even for open-ended and creative generation tasks (Kaddour et al., 2023). However, when generating texts from these probabilistic models, decoding strategy has a prominent effect on the quality of result. Decoding strategies that search for the highest-probability words tend to produce texts that is dull and repetitive (Zhang et al., 2021), while stochastic strategies, which randomly sample from models, generate texts with better human preference (DeLucia et al., 2021; Holtzman et al., 2020). These texts have human-like information distribution, which are viewed as informative and interesting (Meister et al., 2022). Despite this, stochastic strategies are distinct from human. The decoding process is probabilistic and independent, while humans produce language under elaborated cognitive control, especially for creative generation. Creative and nonsense contents are both infrequent during next word prediction that cannot be simply distinguished via sampling strategies (Figure 1). Thus, it is unclear whether LLMs genuinely have creativity during modeling, and whether decoding strategies help.

To answer both of these questions, we evaluate the creativity of LLMs. Specifically, we use an objective semantic measurement, the divergent association task (DAT), which asks models to generate unrelated nouns and compute the pairwise semantic distance between them (Olson et al., 2021). In summary, we make the following contributions:

- Investigate the creativity of LLMs and compare the results with human.

- Explore the effect of decoding strategies on the creative generation of LLMs.

## 2 Measuring Creativity

A direct measure of creativity is relying on experts to judge the creative quality of products. Several

Figure 2: The DAT paradigm and example responses.

studies assessed LLMs' creativity on artistic and scientific creation (Crothers et al., 2023; Park et al., 2023). However, two elements of creativity, novelty and value, are both relative and ambiguous during evaluation. Human ratings are affected by subjective surprise and domain knowledge, and thus differ from each other.

There are other methods based on domain-general cognitive process of creativity.[2] Divergent thinking, i.e., generating a large variety of solutions from an open-ended point, is an indicator for creative thinking (Beaty et al., 2014). One of the most widely used tasks on divergent thinking is the alternate use task (AUT), which asks participants to generate unusual uses of objects (e.g., a brick) (Guilford, 1964). Previous studies used AUT to measure the creativity of LLMs (Summers-Stay et al., 2023; Haase and Hanel, 2023), but the evaluation is sample-dependent that the scores are various across the selected objects. AUT also relies on humans to rate the creativity of generated uses. Moreover, AUT has the the risk of data leakage that the answers are confounded by the uses recorded in the training data.

Creativity has long been linked to the flexibility of semantic retrieval (Zhang et al., 2023). An alternative to probe creativity is through the structure of semantic memory, which can be automatically and reproducibly assessed (Beaty et al., 2021; Beaty and Johnson, 2021). The DAT, among these methods, is valid and reliable that closely correlates with other metrics of creativity (Olson et al., 2021). Different from measuring semantic similarity as usual, the DAT prompts participants to reject related associations and produce unrelated nouns (Figure 2). Formally, given $n$ words and their word embed-

---

[2]Other components of creativity, such as emotion and motivation, are not considered in this study.

dings $\{\boldsymbol{v_1}, \dots, \boldsymbol{v_n}\}$, the DAT can be calculated as the average cosine distance as follows:

$$\text{DAT} = \frac{100}{n(n-1)} \sum_{\substack{i,j \\ i \neq j}}^{n} \left(1 - \cos\left(\boldsymbol{v_i}, \boldsymbol{v_j}\right)\right) \quad (1)$$

In this study, we apply the DAT to assess the creativity of LLMs, but before that it is necessary to evidence the applicability of this method. Basically, the validity of DAT for humans comes with the bias that humans retrieve semantics exploiting their semantic networks. The semantic networks of humans reveal the semantic representations about the world, which are also reflected in the language distribution of human corpus. Thus, LLMs pre-trained on the corpus should exhibit the similar bias. The semantic networks are also needed for LLMs to accomplish general language tasks. Empirically, previous studies showed that language models have similar patterns of semantic activations with humans (Lake and Murphy, 2020; Digutsch and Kosinski, 2023). Additionally, considering these studies assessing the semantic activations differently from the present study, we provide another analysis to validate the DAT for LLMs. It is noteworthy that possessing similar semantic networks is not equivalent to being equally creative. Although the semantic networks of humans are roughly consistent, the ability to produce remote associations is challenging and largely varies among humans.

## 3 Experiment Setup

### 3.1 Models

We studied five recent LLMs with different sizes, including GPT-4 (OpenAI, 2023) and GPT-3.5-Turbo (OpenAI, 2022) from OpenAI, Oasst-Llama-30B (Köpf et al., 2023) and Vicuna-13B (Chiang et al., 2023) fine-tuned from Llama (Touvron et al., 2023), and ChatGLM-6B based on GLM (Du et al., 2022).[3] GPT-4 and GPT-3.5-Turbo have advanced performance through pre-train and RLHF (Ouyang et al., 2022), while other models are trained by fine-tuning foundation models on collected instructions.

### 3.2 Decoding strategy

For deterministic algorithms, we use greedy search that choose the most probable token at each decod-

ing step. For stochastic algorithms, we use top-$p$ sampling (Holtzman et al., 2020) that limit the sampling space to the top-$p$ most likely tokens at each decoding step, truncating the undesirable tail of distribution. We set $p = 0.9$ with temperature $t = 0.7$ for top-$p$ sampling. Then we adjust different settings of $t$ to study the effect of temperature. For each model, we collect enough samples to ensure the results convergent (Appendix A).

### 3.3 DAT

In DAT, we ask models to generate 10 unrelated nouns. we constrain the models to generate only nouns to isolate the semantical distances from syntactic effects. We use the zero-shot prompt in Figure 2 that is consistent with the study for humans (Olson et al., 2021). We filter out the answers with invalid words, e.g., verbs. Then we select the first seven valid words that models provide, and compute the DAT score via Eq. (1).[4]

We use GLoVe (Pennington et al., 2014) to calculate semantic distance (Figure 3a). In previous studies which also used semantic space to evaluate creativity, GLoVe was proved to be effective (Beaty and Johnson, 2020). We have also experimented Word2Vec (Mikolov et al., 2013) and Fasttext (Bojanowski et al., 2016), and found results similar (with the correlation coefficient of 0.82 and 0.91 respectively).

The word vectors also encode word frequency that rare words have unstable semantic distance for the lack of training (Schnabel et al., 2015). Thus, we also compute the average surprisal (negative log word frequency) to study this potential effect.

To compare the result of models with humans, we use the data collected on 8572 participants.[5] We also randomly sample nouns from Wordnet (Miller, 1995) as a situation without the bias of semantic network.

### 3.4 Validating DAT

As mentioned in Section 2, we set two additional prompts for comparison: (1) **Base**: write 10 nouns, and (2) **Random**: write 10 nouns randomly. We hypothesize that LLMs generate semantically associated words if not instructed, but can also have divergent associations under the DAT prompt.

---

[3]Specifically, we use the following versions: gpt-4-0314, gpt-3.5-turbo-0301, oasst-sft-7-llama-30b, Vicuna-13b-delta-v1.1 and chatglm-6b-v1.0.

[4]The number of seven is consistent with Olson et al. 2021 because most answers have at least seven valid words, and results are stable using over seven words.

[5]The data and code of DAT for human is available at https://osf.io/vjazn/.

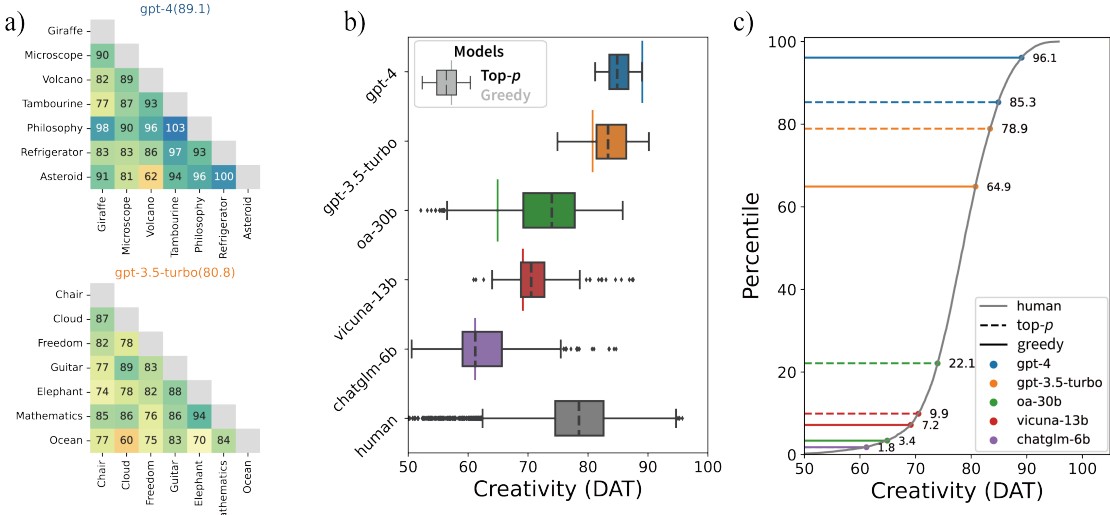

Figure 3: The DAT for humans and models. (a) The distance matrix of words generated by GPT-4 and GPT-3.5-turbo. The average distance is defined as the DAT. (b) The DAT of models and human. (c) The percentile of models' DAT against human results.

## 4 Result

The DAT results are shown in Figure 3. Using greedy search, GPT-4 achieves the highest DAT of 89.1, surpassing 96.1% of humans, and GPT-3.5-Turbo attains a DAT of 80.8 that is above the average human-level (Figure 3c). Other models perform less well with lower DAT, which are roughly proportional to the size of models. When using top-$p$ sampling, models other than GPT-4 are capable of getting the DAT much higher than greedy search, but they also become unstable that probably generate answers with low DAT scores (Figure 3b).

We also report the relation between the DAT and surprisal in Figure 4.[6] Theoretically, surprisal is corresponding to the novelty that is an element of creativity, and the original DAT metric including word frequency effect is valid for human as well (Olson et al., 2021). But as mentioned in Section 3.3, word frequency might be a confounding variable when calculating semantic distance. Indeed, we find two variables highly relevant for human and random baselines (also see Appendix B). For models, the results of top-$p$ sampling have homogeneous slopes with two baselines, but their intercepts and surprisal distributions are different. GPT-4 and GPT-3.5-Turbo exceed the average human DAT under the same surprisal, while other models fall short. Despite Vicuna-13B

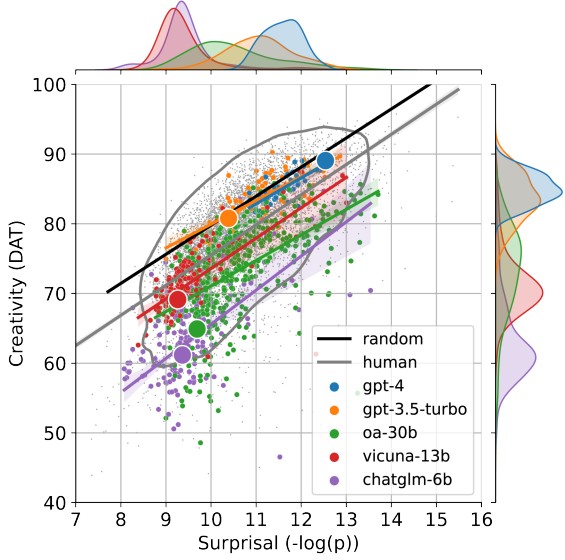

Figure 4: Relationship between the DAT and surprisal. Rimmed points show the results of greedy search. The contour indicates the 95% confidence interval of humans.

and Chatglm-6B have similar distributions of surprisal, the former generates words more divergently. Oasst-Llama-30B defeats Vicuna-13B on the DAT, but this might be explained by the capacity or tendency to generate rarer words. To clarify this effect, we control the surprisal for DAT in Appendix D. The results are similar that GPT-4 and GPT-3.5-Turbo outperform average human performance, but the superiority of GPT-4 is attenuated.

---

[6]The DAT and surprisal for more LLMs are shown in Appendix C

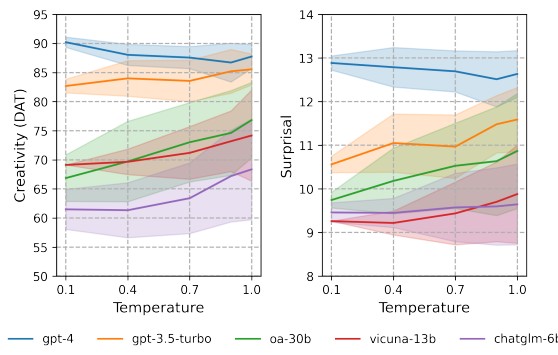

Figure 5: Effect of temperature tuning. The bands indicate the standard deviations.

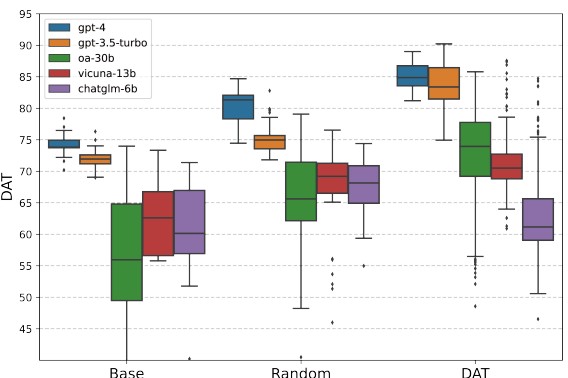

Figure 6: The DAT scores of 3 conditions.

We further investigate the effect of temperature (Figure 5). Temperature is a widely deployed parameter for creative or original generation in practice that high temperature skews the distribution shape towards low-probability tokens (Ackley et al., 1985; Fan et al., 2018). We vary the temperature from 0.1 to 1 and found its positive effect for models except GPT-4. However, the effect is limited, and high temperature will also bring instability and produce invalid answers. As for GPT-4, high-probability tokens are well aligned with high-quality answers.

In the relationship between the DAT and surprisal, we find a naive algorithm that samples nouns from Wordnet can outperform the majority of humans and models (Figure 4). It is because the algorithm has no constrains of the language distribution, which also means it can barely accomplish general language tasks. Although LLMs have exhibited striking mastery of natural language, we wonder whether they process the semantics differently with humans and the DAT test is accordingly invalid as well. Thus, we compare the DAT with Base and Random conditions (figure 6). We show that if not instructed, LLMs tend to produce more related words. When instructed with the prompts of Random and the DAT, LLMs can modulate the language distributions to be more divergent.

These results indicate that LLMs have the potential to generate divergent content with instruction, but with the flaw of not inhibiting common words. Stochastic decoding strategy is helpful for promoting remote association, but since the creative and nonsense content are both infrequent, it cannot accurately produce high-quality content.[7]

However, advanced LLMs show the implicit control of modulating the probability distribution and stably generating divergent answers. Stochastic decoding strategy may even degrade performances for introduced randomness.

## 5 Conclusion

In this work, we provide a creativity evaluation using a divergent semantic task. This task reveals distinguishable results across various LLMs and decoding strategies. We find GPT-4 demonstrates advanced human-level creativity stably, while GPT-3.5-turbo exceeds the average human level. For decoding methods, stochastic decoding strategy is effective but not enough for creative generation.

## Limitation

Creativity is a deeply debated concept. We selectively evaluate the "little-C" (creativity in everyday life) that is a general capacity, instead of the "big-C" (marvelous creative product) which is rare even for human. Measuring creativity is also controversial that requires evaluations from multiple perspectives, principles, and analysis. Thus, the results of this study cannot be directly generalized to all language generation tasks. We also limited the range of this study within self-contained creativity, whereas another crucial aspect of AI's creativity is human-AI co-creation. We leave these for future work.

## Acknowledgement

We thank Mark Sun and the anonymous reviews for their thoughtful helps and suggestions. This work was supported by STI2030-Major Projects 2021ZD0204105 and Fundamental Research Funds for the Central Universities 226-2023-00091.

---

[7]Previous psychological researches reported similar result that mild imperfection of attention is related to higher creativity (Abraham, 2014).

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

## A    Selecting sample size for each model

Figure 7 shows the sample size to generate stable results using top-$p$ sampling for each model. With confidence coefficient $\alpha = 0.05$, standard deviation $\hat{\sigma}$ and error $\epsilon = 1$, we choose $N > (\lambda_\alpha \times \hat{\sigma}/\epsilon)^2 = 1.96^2 \times \hat{\sigma}^2$. We find larger models (GPT-4 and GPT-3.5-Turbo) generate answers more stably.

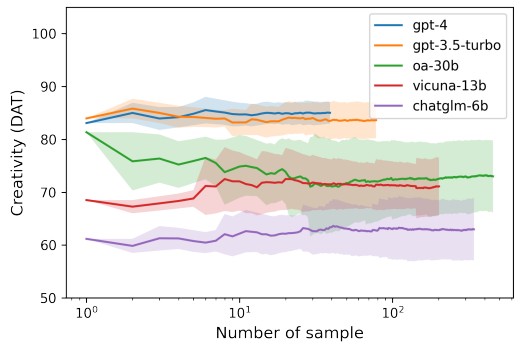

Figure 7: Results of the DAT across different sample sizes. The bands indicate the standard deviations.

## B    Results of the DAT and surprisal on human and random and baselines

Figure 8 shows the results of the DAT and surprisal on human and random baselines. We find positive relationship between surprisal and the DAT. Random baseline is a strong baseline that has maximal remote association (uniform distribution) despite without inhibition. Even so, we find some people surpass random baseline and approach ceiling DAT at specific section of surprisal.

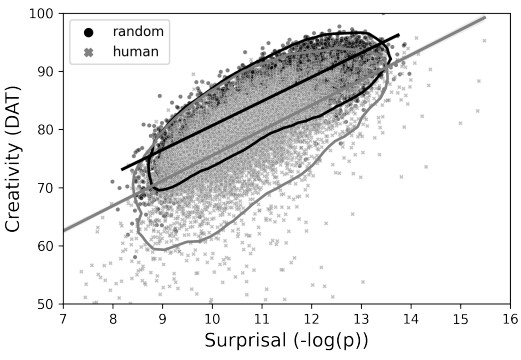

Figure 8: Results of the DAT and surprisal on human and random baseline. Contours indicate the 95% confidence intervals.

## C    Result of the DAT and surprisal for more models using greedy search

Figure 9 shows the results of the DAT and surprisal for more LLMs using greedy search.

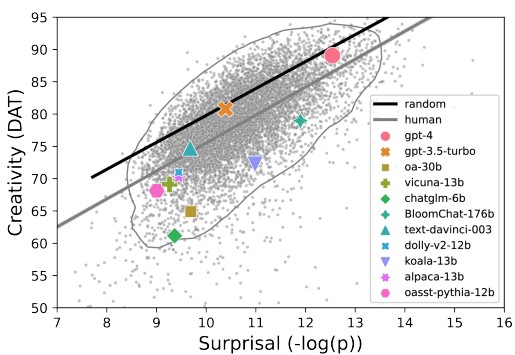

Figure 9: Results of the DAT and surprisal for more LLMs. The contour indicate the 95% confidence interval.

## D    Controlling surprisal as a confounding variable

Considering the potential influence of word frequency on measuring semantic distance, we control surprisal (Figure 10 ). The results are similar as before that GPT-4 and GPT-3.5-Turbo outperform average human performance, while other models are below average human level. However, GPT-4 as well as Oasst-Llama-30B lose their superiorities because their high DAT scores partially depend on generating rarer words.

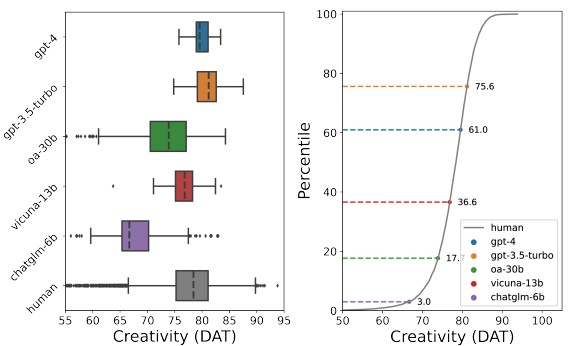

Figure 10: Results of the DAT when controlling surprisal as a confounding variable.