# OpenReview forum: "Probing the “Creativity” of Large Language Models:  Can models produce divergent semantic association?"
_EMNLP/2023/Conference — EMNLP 2023 Findings_

### Official Review · Reviewer_qcAx · 2023-07-28

**Soundness:** 3

**Excitement:**

3: Ambivalent: It has merits (e.g., it reports state-of-the-art results, the idea is nice), but there are key weaknesses (e.g., it describes incremental work), and it can significantly benefit from another round of revision. However, I won't object to accepting it if my co-reviewers champion it.

**Missing References:**

-

**Paper Topic And Main Contributions:**

The paper investigates the *creativity* of Language Models, in a quantifiable way. A common critique of LMs is that while they are highly adapt at imitating human language use, they still severely lack in creativity.

Central to the paper is the Divergent Association Task (DAT), a recently introduced task (Olson et al., 2021) which has been shown to correlate strongly with creativity in humans. The task asks to generate a list of 10 nouns that are as irrelevant as possible from each other.

The authors consider various recent LLMs of increasing sizes, and use the cosine similarity between the generated nouns using GLoVe embeddings to compute the divergence score of the DAT task.

The authors show that their largest model, GPT-4, scores better on the DAT task than human performance, and that creativity improves with model size. Other factors that are investigated is the connection between the DAT score and model surprisal (positive correlation), and the impact of the decoding strategy (greedy vs. stochastic) and the influence of temperature on decoding.

**Questions For The Authors:**

- How can the (cosine) distance between words be larger than 100? (Fig. 3a, Philosophy-Tambourine).
- How do you ensure/validate that the generated words are all nouns and not other types of words?

**Reasons To Accept:**

- The paper provides a focused analysis of creativity in LLMs, and addresses various interesting factors related to creativity (surprisal, temperature).
- The authors propose a simple and computationally efficient way of assessing creativity in LLMs, leveraging findings from creativity assessment in human language processing.

**Reasons To Reject:**

The paper makes various assumptions that are not adequately addressed:
- The findings for creativity found in humans where the DAT task predicts human creativity does not necessarily transfer to LLMs; taking this as a given skips the step of validating the finding of Olson et al. (2021) for the domain of LLMs.
- Cosine similarity has traditionally been shown to correlate with semantic relatedness, but its one-dimensional nature makes it suboptimal to measure semantic similarity in isolation. E.g., by requesting to only predict nouns we are already restricting token predictions to come from a particular "region" in the embedding space. Since cosine similarity does not differentiate between semantic relatedness and other linguistic dimensions (syntact/morphological/etc.), a model could "cheat" by generating words that are not only semantically unrelated, but also syntactically.
- DAT is computed only using the GLoVe cosine distances, but the impact of this choice is not addressed. GLoVe vectors should not be taken as a gold truth for semantic similarity.

**Reproducibility:**

4: Could mostly reproduce the results, but there may be some variation because of sample variance or minor variations in their interpretation of the protocol or method.

**Reviewer Confidence:**

3: Pretty sure, but there's a chance I missed something. Although I have a good feel for this area in general, I did not carefully check the paper's details, e.g., the math, experimental design, or novelty.

**Typos Grammar Style And Presentation Improvements:**

The title is ungrammatical, it should be "Probing" instead of "Probe"

- L031: "Reasoning as a core aspect of intelligence [..] has been found from LLMs" reads odd
- L045: What does "Using creativity for self-iteration" imply? This is too vague to me.
- I fail to see how Figure 1 exemplifies that "inhibiting common ideas emphasizes top-down attention".
- L068: "program" -> "programming"? "Computer programs"?
- L115: "process" -> "processes"
- In general the writing would benefit from another pass, in a lot of cases singular nouns are missing a determiner (or they should be pluralised).

- The lines in Figure 4 are quite hard to discern with all the scatter info in there as well. Increasing the opacity of the scatter points could already help here, to make the lines stand out more.

---

> ### Author Rebuttal · Authors · 2023-08-27
>
> We thank you for your helpful comments and careful corrections about the typos.
>
> - **The applicability of the DAT paradigm**. The first concern is that DAT might not transfer to LLMs without a validation. On the one hand, as there is no criterion about the creativity of LLMs, we cannot strictly validate it. However, we compared the results with a recent study [1], which uses AUT to evaluate the creativity of LLMs, and found similar results. It shows GPT-4 approaches the max human score, while GPT-3 achieves average human score, followed by Alpaca and other models. We appeal more related works for comprehensive validation and evaluation. On the other hand, previous works indicate that LLMs possess similar patterns of semantic activations with humans [3-4], which means LLMs has a similar semantic bias. Based on this assumption, we consider DAT as an appropriate method to transfer from humans to LLMs. In the final version, we would supply these arguments into the sections of *Measuring Creativity* and *Result*.
> - **The one-dimensional nature of cosine similarity**. As you said, a model could "cheat" by generating words that are not only semantically unrelated, but also syntactically. Thus, we constrain the models to generate only nouns to isolate the semantical unrelatedness. And you also concern about this requirement would restrict tokens to come from a particular “region”.  However, we note that using the semantic space of nouns is sufficient to distinguish the creativity among models and humans. It is effective, though considering words other than nouns is more comprehensive. We would refine the method description in the final version.
> - **DAT is computed only using the GLoVe cosine distances**. In previous studies which also used semantic space to evaluate creativity, GLoVe was proved to be useful [4-5]. We have also experimented various word vectors, e.g., Word2Vec and Fasttext, and found similar results (with the correlation coefficient of 0.82 and 0.91 respectively). In the final version, we would detail the comparing of word vectors to indicate this impact in the appendix.
>
> In response to your more detailed questions,
>
> - **Q1**: How can the (cosine) distance between words be larger than 100? (Fig. 3a, Philosophy-Tambourine).
>
>     **A1**: The cosine distance is calculated as 1-cos(word_1,word_2), and we multiply it by 100, thus the distance ranges from 0 to 200. In practice, scores commonly range from 65 to 90 and rarely exceed 100.
>
> - **Q2**: How do you ensure/validate that the generated words are all nouns and not other types of words?
>
>     **A2**: We add the requirement in the prompt, and the models generally follow our instruction. In the analysis, we also use wordnet to validate whether the answers are nouns.
>
> - **Q3**: What does "Using creativity for self-iteration" imply? This is too vague to me.
>
>     **A3**: A recent study shows that the use of model-generated content in training causes irreversible defects in the resulting models, where tails of the original content distribution disappear [6]. They refer to it as *model collapse*, i.e. the vanishment of the tails of the original content distribution would make LLMs converge to a limited set of outputs. In other words, LLMs at present always generate common tokens, while the lack of creative generation would pollute the corpus. Thus, if models can generate content creatively, this problem can be largely settled.
>
> - **Q4**: I fail to see how Figure 1 exemplifies that "inhibiting common ideas emphasizes top-down attention".
>
>     **A4**: We are sorry that we use figure 1 to show the remote association and inhibition (a process of cogtive control) in the creative generation, while the top-down control of attention is from the cited dual-process theory. We would like to compare this with the sampling strategies for LLMs. But as we have mentioned this point in the next paragraph, we will delete it in the current paragraph.
>
>
> We would refine the article according to your useful suggestions. We hope that our responses and updates can be carefully considered.
>
> **Reference**:
>
> [1] Haase, J., & Hanel, P.H. (2023). Artificial muses: Generative Artificial Intelligence Chatbots Have Risen to Human-Level Creativity. *ArXiv, abs/2303.12003*. *https://doi.org/10.48550/arXiv.2303.12003*
>
> [2] Jan Digutsch and Michal Kosinski. (2023). Overlap in meaning is a stronger predictor of semantic activation in GPT-3 than in humans. Scientific Reports, 13(1):5035. https://doi.org/10.1038/s41598-023-32248-6
>
> [3] Lake, B. M. & Murphy, G. L. (2021). Word meaning in minds and machines. Psychol. Rev. https://doi.org/10.1037/rev0000297
>
> [4] Jay A. Olson, Johnny Nahas, Denis Chmoulevitch, Simon J. Cropper, and Margaret E. Webb. (2021). Naming unrelated words predicts creativity. Proceedings of the National Academy of Sciences, 118(25). https://doi.org/10.1073/pnas.2022340118
>
> [5] Beaty, R.E., Johnson, D.R. (2021). Automating creativity assessment with *SemDis*: An open platform for computing semantic distance. *Behav Res* **53**, 757–780. https://doi.org/10.3758/s13428-020-01453-w.
>
> [6] Ilia Shumailov, Zakhar Shumaylov, Yiren Zhao, Yarin Gal, Nicolas Papernot, and Ross Anderson. (2023). The curse of recursion: Training on generated data makes models forget. *ArXiv, abs/2307.10169*. https://doi.org/10.48550/arXiv.2307.10169

---

### Official Review · Reviewer_hFka · 2023-08-01

**Soundness:** 4

**Excitement:**

4: Strong: This paper deepens the understanding of some phenomenon or lowers the barriers to an existing research direction.

**Paper Topic And Main Contributions:**

The study probes a host of LLMs for their ability to generate divergent associations using the DAT. The study further assesses the impact of the temperature parameter on the average semantic similarity of generated responses. In doing so, the study controls for surprisal, since low frequency words may have less stable semantic representations.

In general, I find the paper misses the mark substantially in probing LLMs for creative thinking, as evidenced by the performance of a random WordNet-based baseline, which succeeds at the task but can hardly be qualified as creative. Maybe I'm wrong here, but the DAT is validated with humans, who come with their biases about retrieving associations exploiting the semantic network and its structure. Showing divergent associations in a model on a task validated and calibrated with humans without first assessing whether that model replicates the same biases humans show when producing free associations does not show creativity at all. So, in order for these results to be taken seriously I think it is paramount to show that the same models exhibit foraging patterns when producing free associations (e.g., when ask to generate as many words starting with, say, the latter b, humans typically start from words from a same semantic field, deplete what they can think of then move to a different semantic field, deplete it, and so on, or when they are asked to generate words for animals they might begin from farm animals, then move to birds, then fish, and so on...). the DAT measures creativity given strong assumptions about the underlying semantic space and strategies to query it, but the paper does not do enough in showing that models match this. Without this bit, a simple random algorithm exploiting a taxonomy wins at being creative, which makes little sense.

-----------------------------------------
The extra analyses provided in the rebuttal do improve the credibility of the paper but I still think a crucial aspect is missing, which I articulated in the comment to the rebuttal.

-----------------------------------------
The second set of extra analyses now more convincingly show that if not explicitly instructed, LLMs will retrieve related concepts but some can overcome this bias if asked to retrieve unrelated items.

**Questions For The Authors:**

Figure 1 is rather unclear to me: what should it show?

page 2, right column, mid: you mention AUT is not a good evaluation as it is sample-dependent, subjective, and risks data leakage: can you substantiate this criticism more?

page 3, right column, top: how are valid words determined?

you operationalise surprisal as the inverse of token frequency, but the second word is generated in the context of the first, and so on, so why not also considering surprisal given the words already generated?

page 4, left column, top: probably generate low quality answers: why probably? what do you mean?

**Reasons To Accept:**

The question is definitely timely and pressing, careful studies on the matter are needed and likely to get us new knowledge.

**Reasons To Reject:**

See above: I think the paper does not show what it purports to show due to a conceptually flawed experimental design.

-----------------------------------
I think the analyses provided are now sufficiently informative. The rhetoric should probably be toned down a bit about this being creativity rather than a narrow aspect of it, but I do see the pros of having a catchy title. The presentations should also be improved.

**Reproducibility:**

5: Could easily reproduce the results.

**Reviewer Confidence:**

4: Quite sure. I tried to check the important points carefully. It's unlikely, though conceivable, that I missed something that should affect my ratings.

**Typos Grammar Style And Presentation Improvements:**

The paper is not very well written and would benefit from a thorough copy-editing.
-----------------------
This likely still applies, but it's hard to judge whether it's been addressed from the rebuttals.

---

> ### Author Rebuttal · Authors · 2023-08-27
>
> We appreciate your constructive and thoughtful feedback. You raised an important concern, and below are some explanations about the remarks.
>
> - The main controversy is the applicability of the DAT paradigm on LLMs. We agree with your exact description about the biases of humans that retrieves associations by exploiting the semantic networks, and we would like to show the same biases for LLMs. Theoretically, the semantic networks of humans reveal the semantic representations about the world, which are also reflected in the corpus. Thus, LLMs pretrained on the corpus exhibit the similar bias. In other words, LLMs learn a language distribution from the corpus, and therefore derive similar word associations with humans. Empirically, Digutsch and Kosinski [1] revealed LLM’s similar patterns of semantic activations with humans that is mentioned in page2, line 146. More evidences can be found here [2-3]. These studies have shown the applicability and challenge of the DAT paradigm on LLMs that LLMs struggle to generate unrelated words under the constraint of language distribution. Our results also verified this restriction. In the final version, we would supply these arguments into Section Measuring Creativity.
> - The random baseline, as you said, does not have such a bias. That means an extreme flexibility of semantic retrieval [4], but does not mean all aspects of creativity. In the section of measuring creativity, we indicate that divergent thinking is an indicator for creative thinking, while creativity also need convergent thinking that is choosing the best answer to a problem based on divergent thinking. The DAT reflects an important aspect of creativity, and we appeal for more comprehensive evaluations.
> - Since the random baseline does not have such a bias, it seems meaningless to set such a random condition. But we would like to emphasize the importance of randomness towards creativity, which is also the reason we assess the impact of decoding strategies. For human, creativity roots in the incidental connections that seemly different things can be related. For machines, evidences from AI painting and NLG underline the effect of random seed and sampling strategies. Indeed, we found some models leverage sampling strategies to improve DAT scores. Even so, randomness is not perfect, for it cannot reject related words. The high DAT score of random baseline comes from the property of high dimensional vector space that two random words are generally distant. However, we can see many people and some models approach or exceed random baseline, in spite of the constraint on flexibility.
> - To make these arguments more convincible, we conduct two brief supplementary experiments:
>     1. **Random generation of LLMs**. We ask LLMs to generate nouns randomly and find models cannot literally do that like a random baseline. Instead, regardless of controlling surprisal or not, they get DAT scores worse than under the original DAT prompt, let alone the random baseline. The results indicate LLMs have the bias to generate related tokens, i.e., they have limited remote association.
>     2. **Choosing unrelated words from a set of given words**. We further control this bias that we sample 30 nouns words first and ask models to choose 10 nouns that are as unrelated as possible with each other. As the words are given, there are no difference on the retrieval flexibility. We find that when exposed to a set of words, LLMs can perform better than randomly choosing 10 nouns from 30 given nouns, especially for GPT-3.5-turbo and GPT-4. The results show that LLMs have the capacity of recognizing unrelated words, i.e., the ability of cognitive control in the theory framework of creativity.
>
>     As creativity is a integrated process, we tend to use the DAT score as a global metric. However, these supplementary results detail the mechanisms of DAT task for LLMs. We would like to supply these results in the final version into the result section or appendix.
>
> In response to your more detailed questions,
>
> - **Q1**: Figure 1 is rather unclear to me: what should it show?
>
>     **A1**: Figure 1 shows the theory framework of creative generation. Creative generation differs from the general language generation that the content should be novel and valuable. Considering language generation as decoding from a conditional probability distribution, in creative generation, we should inhibit the probability of common tokens and enhance the probability of creative tokens. Based on this framework, we investigate the creativity of LLMs that whether these models can reject related words and generate unrelated words.
>
> - **Q2**: page 2, right column, mid: you mention AUT is not a good evaluation as it is sample-dependent, subjective, and risks data leakage: can you substantiate this criticism more?
>
>     **A2**: AUT asks participants to generate unusual use of objects, and it has a standard questionnaire. The number of objects is generally 3-5, while the scores are various across the objects selected. What’s more, AUT relies on humans to rate the creativity of generated unusual uses, which is subjective. Lastly, since AUT is a classic test with standard scale, there are many answers on the web, which has the risk of data leakage.
>
> - **Q3**: page 3, right column, top: how are valid words determined?
>
>     **A3**: Valid words refers to the words that satisfy the requirement in the prompt, i.e., single words, only nouns, and no proper nouns.
> - **Q4**: you operationalise surprisal as the inverse of token frequency, but the second word is generated in the context of the first, and so on, so why not also considering surprisal given the words already generated?
>
>     **A4**: We use the token frequency here in order to check the effect of independent word frequency, which might be confusing. We wonder if you want to confirm that, since DAT prompts models to generate unrelated words, the surprisal given the words already generated should be large. However, it is hard to calculate the surprisal because noun sequences are not natural sentences, and we cannot define a standard language distribution. Instead, we use the semantic distance to quantify the unrelatedness.
> - **Q5**: page 4, left column, top: probably generate low quality answers: why probably? what do you mean?
>
>     **A5**: The low-quality answers refer to the answers with DAT scores lower than greedy search. We show that sampling strategies generate unstable results because these strategies would choose some related words with low probability that have been rejected by LLMs.
>
> We would refine the article in the final version according to your useful suggestions. We hope that our responses and updates can be carefully considered.
>
> ### **reference**:
>
> [1] Jan Digutsch and Michal Kosinski. (2023). Overlap in meaning is a stronger predictor of semantic activation in GPT-3 than in humans. Scientific Reports, 13(1):5035. https://doi.org/10.1038/s41598-023-32248-6
>
> [2] Lake, B. M. & Murphy, G. L. (2021). Word meaning in minds and machines. Psychol. Rev. https://doi.org/10.1037/rev0000297
>
> [3] Lenci, A., Sahlgren, M., Jeuniaux, P. *et al.* A comparative evaluation and analysis of three generations of Distributional Semantic Models. *Lang Resources & Evaluation* **56**, 1269–1313 (2022). https://doi.org/10.1007/s10579-021-09575-z
>
> [4] Jingyi Zhang, Kaixiang Zhuang, Jiangzhou Sun, Cheng Liu, Li Fan, Xueyang Wang, Jing Gu, and Jiang Qiu. (2023). Retrieval flexibility links to creativity: evidence from computational linguistic measure. Cerebral Cortex, 33(8):4964–4976. https://doi.org/10.1093/cercor/bhac392

---

### Official Review · Reviewer_9M5g · 2023-08-05

**Soundness:** 3

**Excitement:**

3: Ambivalent: It has merits (e.g., it reports state-of-the-art results, the idea is nice), but there are key weaknesses (e.g., it describes incremental work), and it can significantly benefit from another round of revision. However, I won't object to accepting it if my co-reviewers champion it.

**Paper Topic And Main Contributions:**

The authors try to quantify the creativity of LLMs through an  objective semantic measurement, the divergent association task (DAT). They further explore the effect of decoding strategies through decoding strategies.

**Questions For The Authors:**

A. What is the reason to use Glove to to calculate semantic distance in DAT?

B. Have you thought about integrating surprisal score with DAT score for measuring creativity?

**Reasons To Accept:**

The paper discusses an interesting take on creativity of LLMs.

**Reasons To Reject:**

The authors specifically looks at single nouns but do not adequately justify why combinations of multiple words (specifically creativity in language is tightly related to the ability to put together two seemingly different words that could mean something interesting, taken together) were not taken into consideration or why words other than nouns were not considered. These would be necessary points to be discussed in the limitations section.

**Reproducibility:**

4: Could mostly reproduce the results, but there may be some variation because of sample variance or minor variations in their interpretation of the protocol or method.

**Reviewer Confidence:**

4: Quite sure. I tried to check the important points carefully. It's unlikely, though conceivable, that I missed something that should affect my ratings.

**Typos Grammar Style And Presentation Improvements:**

The paper could use a complete revision to better structure the sentence and overall structure of the article. (e.g., Word frequency, however, is also encoded in the embedding vectors that rare words might have unstable semantic distances --> sentences like these are harder to understand)

---

> ### Author Rebuttal · Authors · 2023-08-27
>
> We thank you for your helpful comments. As you said, creativity in language is related to the ability to put together two seemingly different words that are actually related and could mean something interesting. But measuring the combinations of multiple words directly is challenging that the interesting compositional meaning is hard to define and assess in a standard way. In DAT, we potentially require language models to recognize and reject these word pairs that are related in some aspects. As for the words other than nouns, we consider the syntactic factors might influence the semantic distances, which could be confusing. We note that using the semantic space of only nouns is sufficient to distinguish the creativity among models and humans. However, considering word combinations and words other than nouns is undoubtedly desirable. We would refine the method description and discuss these points in the limitations section.
>
> In response to your more detailed questions,
>
> - Q1: What is the reason to use Glove to to calculate semantic distance in DAT?
>
>     A1: Since the DAT task is not contextual, the semantic distance should be calculated using static word vectors, while Glove provides well-constructed static vectors for word representations. In the study that introduced DAT for humans [1], the authors have demonstrated that DAT scores calculated by Glove correlate strongly with other creativity metrics. In other studies which used semantic space to evaluate creativity, Glove was proved to be useful as well [2]. Thus, we kept the same experimental setup here. We have also experimented various word vectors, e.g., Word2Vec and Fasttext, and found similar results (with the correlation coefficient of 0.82 and 0.91 respectively). We would discuss the impact of choosing Glove in the appendix.
>
> - Q2: Have you thought about integrating surprisal score with DAT score for measuring creativity?
>
>     A2: Thank you for your suggestion. Surprisal score reveals the ability to generate low-frequency words, which is relevant to creativity. For humans, we do find surprisal score has a significant correlation with other creativity metrics. But as evidenced in the experiment, we find DAT score correlated with surprisal score closely, and integrating surprisal with DAT do not improve the predictive power for other metrics. In other words, the original DAT score has already integrates the effect of surprisal.
>
>
> **Reference**:
>
> [1] Jay A. Olson, Johnny Nahas, Denis Chmoulevitch, Simon J. Cropper, and Margaret E. Webb. (2021). Naming unrelated words predicts creativity. Proceedings of the National Academy of Sciences, 118(25). https://doi.org/10.1073/pnas.2022340118
>
> [2] Beaty, R.E., Johnson, D.R. (2021). Automating creativity assessment with *SemDis*: An open platform for computing semantic distance. *Behav Res* **53**, 757–780. https://doi.org/10.3758/s13428-020-01453-w.

---

### Meta-Review · Area_Chair_YNyX · 2023-09-15

**Recommendation:** 4

**Metareview:**

This paper aims to probe the creativity of LLMs, operationalized in terms of divergent semantic association.
The reviewers agree that this is a timely topic, and provided a relatively positive assessment.

The scores were relatively consistent, in the medium to high range:  3,4,3 for both soundness and excitement. Some of the concerns raised in the original reviews were addressed in the discussion period, leading to increases in the scores.

Distilling the reviewers, the following major strengths and weaknesses were noted.

Strengths:

- The paper was judged as studying an interesting and timely question, in an interesting and intuitive way.

Weaknesses:

- paper specifically considered single nouns (R1)
- R2 (hFka) originally considered the paper insufficient, but was convinced by additions made during the discussion period.
- R3 originally found various unaddressed assumptions (validating DAT task for LLMs, concerns about cosine distance, focusing on GLoVe). After the author response, they found some of their concerns to have been addressed.

In conclusion, this paper was judged to be good or strong in terms of soundness, and scored "ambivalent" or even "strong" in terms of excitement.

---

### Decision · Program_Chairs · 2023-10-07

**Decision:**

Accept-Findings

**Comment:**

This paper aims to probe the creativity of LLMs, operationalized in terms of divergent semantic association.
The reviewers agree that this is a timely topic, and provided a relatively positive assessment.

The scores were relatively consistent, in the medium to high range:  3,4,3 for both soundness and excitement. Some of the concerns raised in the original reviews were addressed in the discussion period, leading to increases in the scores.

Distilling the reviewers, the following major strengths and weaknesses were noted.

Strengths:

- The paper was judged as studying an interesting and timely question, in an interesting and intuitive way.

Weaknesses:

- paper specifically considered single nouns (R1)
- R2 (hFka) originally considered the paper insufficient, but was convinced by additions made during the discussion period.
- R3 originally found various unaddressed assumptions (validating DAT task for LLMs, concerns about cosine distance, focusing on GLoVe). After the author response, they found some of their concerns to have been addressed.

In conclusion, this paper was judged to be good or strong in terms of soundness, and scored "ambivalent" or even "strong" in terms of excitement.